# Improved MTF Measurement of Medical Flat-Panel Detectors Based on a Slit Model

**DOI:** 10.3390/s25051341

**Published:** 2025-02-22

**Authors:** Haiyang Zhang, Zhiyong Ji

**Affiliations:** College of Engineering Science and Technology, Shanghai Ocean University, Shanghai 201306, China; 19946220047@163.com

**Keywords:** modulation transfer function (MTF), oversampling, flat-panel detector, slit model

## Abstract

In the development, evaluation, and application of medical flat-panel detectors, the modulation transfer function (MTF) is crucial, as it reflects the device’s ability to restore detailed information. Medical flat-panel detectors encompass both image data acquisition and digitization processes, and detectors with varying pixel sizes exhibit differing capabilities for observing details. Accurately quantifying MTF is a critical challenge. The complexity of MTF calculation, combined with unclear principles and details, may result in erroneous outcomes, thereby misleading research and decision-making processes. This paper presents an improved MTF oversampling method based on the slit model. MTF testing is conducted under various sample conditions and using different focal spot diameters of the X-ray tube to analyze the impact of focal spot size. High-precision tungsten plates and fixtures are designed and fabricated, and MTF results with varying line spread function (LSF) sampling intervals are compared. The results demonstrate that the improved slit model offers distinct advantages, with MTF measurements achieving 92.4% of the ideal value. Compared to traditional tungsten edge and point (aperture) model testing methods, the accuracy of the proposed method is improved by 5–13%. The optimal sampling interval is approximately 1/29 of the pixel pitch, offering a more accurate method for evaluating detector performance.

## 1. Introduction

In the domain of research and application of medical flat-panel detectors, the modulation transfer function (MTF) plays a pivotal role. It is imperative for accurately assessing detector performance and ensuring the quality of medical imaging [1]. The MTF is fundamentally a quantitative description of the attenuation of signal contrast as transmitted by optical or imaging systems. In the research and development process, the performance evaluation and clinical application of medical flat-panel detectors serve as a key metric that precisely reflects the device’s capability to restore fine image detail [2].

In recent years, the development of medical equipment innovation and imaging technology has become an important driving force for improving diagnostic accuracy and medical efficiency. For example, some studies have explored the strategy of the innovative selection of medical devices, emphasizing the impact of technological breakthroughs on product performance optimization [3]. In addition, some studies have analyzed how medical manufacturing enterprises can improve the efficiency of technological innovation, pointing out that accurate detection and data analysis methods are crucial to the manufacturing process [4]. The MTF measurement improvement scheme proposed in this study is consistent with the core idea of the above research, i.e., by optimizing the imaging measurement method, the performance and reliability of medical imaging equipment are improved, and the overall efficiency of medical device manufacturing and application is improved.

The workflow of medical flat-panel detectors encompasses two essential steps: image data acquisition and digitization. During this process, variations in pixel sizes result in significant differences in the detector’s ability to capture fine details [5]. Consequently, the precise quantification of MTF becomes a critical concern. The MTF calculation process involves intricate principles and detailed procedures. Errors in these calculations can potentially misdirect the development trajectory of the detectors and related decision-making processes, severely impacting research efficiency and the quality of outcomes.

Oversampling technology plays a crucial role in enhancing the accuracy of MTF measurements. The core principle involves sampling at a rate higher than the Nyquist sampling rate, effectively reducing noise interference and minimizing aliasing effects, thereby achieving more precise data fitting. However, it should be noted that current oversampling methods are not without their limitations. For example, although increasing the sampling frequency may improve accuracy in theory, it can result in an exponential increase in data volume, which can impose substantial demands on storage capacity and computational processing power, thereby significantly increasing hardware costs and processing burdens [6]. Multi-pixel merging strategies can simplify the data processing workflow to a certain extent, but they may compromise spatial resolution, leading to a loss of image detail. While interpolation algorithms can complement certain data under specific conditions, they may introduce new errors when applied to critical areas such as edges, resulting in the distortion of image details and affecting the accuracy of MTF measurements.

In the process of calculating the MTF for a specific detector with a small pixel spacing, it was found that the MTF results were lower than the theoretical expectations or than those of similar products. Further investigation revealed several contributing factors [7]. Among these, the appropriateness of the LSF sampling interval setting is critical, as improper settings fail to accurately capture signal features. The size of the X-ray tube’s focal spot has been found to affect imaging clarity, with larger focal spots potentially leading to image blurring. Scattering phenomena have been shown to interfere with the propagation of light, reducing image contrast. Insufficient processing accuracy of the tungsten plate edges has been identified as a factor in blurry edge imaging, thereby affecting measurement standards. Inadequate adherence of the tungsten plate to the detector surface has been found to result in light transmission loss and deviations.

In response to the aforementioned issues, this paper conducts an in-depth study to improve the MTF oversampling method based on the slit model. A series of MTF testing experiments has been meticulously designed and implemented for diverse samples, thereby ensuring comprehensive coverage of the test scenarios under various X-ray tube focal spot diameters. This encompasses typical cases such as small focal points (0.6 mm [8]), focal points employed in mammography (0.1 mm [9]), and intraoral dental X-ray tubes (0.7 mm [10]). The study systematically analyzes the impact mechanism of focal spot size on MTF. Additionally, high-precision manufacturing processes are used to design and fabricate tungsten plates with higher edge accuracy and custom fixtures for small-pixel detector MTF testing, thereby enhancing measurement accuracy at the hardware level. Furthermore, by delving into the core principles of MTF calculation and sampling, a thorough and detailed comparative analysis of MTF results under different LSF sampling intervals is conducted, successfully identifying the optimal oversampling step for small-pixel detectors. This approach has been validated for its universal applicability to flat-panel detectors with different pixel sizes, providing a more precise and reliable method for evaluating the performance of medical flat-panel detectors. This work represents a substantial advancement in the development and progress of medical imaging technology.

## 2. MTF Testing Method

The MTF (modulation transfer function) is a pivotal parameter employed to assess the performance of imaging systems [11]. It is defined as the ratio of the contrast (or modulation) of the output image to the contrast (or modulation) of the input image, thereby offering a comprehensive reflection of the spatial resolution and contrast of the detector [12]. The MTF value ranges from 0 to 1, with 1 representing the system’s maximum capacity to reproduce imaged objects at different spatial frequencies. A value of 1 indicates that the system can fully reproduce the input image contrast without any loss, while a value of 0 means the system is unable to reproduce the contrast of the input image at all. The MTF evaluates the contrast resolution of a detector at specific spatial frequencies by quantifying the system’s response to different frequency components [13]. The MTF measurement methods have presented various research statuses and challenges. With regard to the current state of research, it is evident that traditional measurement techniques are undergoing continuous refinement. A notable example is the knife-edge method [14], which utilizes interpolation and fluctuation to address the measurement errors caused by angle discrepancies, thereby enhancing the precision requirements. The advent of novel principles and technologies has precipitated the evolution of pioneering methodologies, exemplified by Kalman filtering-based techniques [15], which methodically process the measured line spread function (LSF) to enhance precision. The integration of disparate measurement techniques has emerged as a pivotal paradigm, with methodologies such as the amalgamation of edge methods with other approaches being instrumental in deriving more precise MTF values [16]. Furthermore, specific measurement methods for medical imaging have been explored, including research conducted by Liu Chuanya’s team from the Shan-dong Institute of Medical Imaging [17], who measured the MTF along the z-axis using a 64-slice spiral CT platform, and the research conducted by the team led by Qin Weichang from Siemens (China) Healthcare Systems Group [18], focusing on the z-axis MTF of the 64-slice spiral CT, contributing to the enhancement of resolution characteristic evaluations in CT systems for medical imaging. However, the accuracy of these measurement methods is influenced by system noise and detector performance, with a particularly significant reduction in measurement accuracy at high frequencies. Additionally, the measurement process has stringent requirements for both equipment and environmental conditions, with professional measuring instruments being expensive and requiring stable environments and precise lighting, thus limiting their widespread use. Furthermore, the measurement speed is slow, and the large volume of data collected and the complex computational processing required makes it difficult to meet the need for rapid measurements. Existing methods also have limitations when applied to complex optical imaging systems, rendering them incapable of accurate measurements. The challenges of data processing and analysis are considerable, with different processing methods and parameter selections affecting the results, thus increasing uncertainty.

In practical scenarios, the MTF is influenced by various factors, including the following: 1. pixel size [19]—reduced pixel sizes generally yield higher spatial resolution; however, excessively diminutive pixel sizes can induce crosstalk, resulting in signal confusion or distortion, thereby compromising image quality; 2. scintillator and TFT alignment [20], i.e., the optical properties of the scintillator and the precision of the alignment between the thin-film transistor (TFT) and the detector; 3. electronic design—the quality of the system’s electronic design influences the MTF [21,22,23]; 4. aliasing effect [24], i.e., the aliasing phenomenon resulting from insufficient sampling; 5. spot size and scattering [25], i.e., the size of the spot and the scattering effects of light within the detector. Therefore, although the highest spatial frequency that a detector can theoretically achieve (referred to as the Nyquist frequency, or P) can be calculated, the actual resolution of the detector, in terms of line pairs per millimeter (or spatial frequency), typically falls short of this theoretical limit. This discrepancy arises due to the influence of the aforementioned factors on the MTF of real detectors, thereby hindering the system’s performance from attaining its theoretical value.

It is generally accepted that traditional DR detectors feature relatively large pixel sizes (typically > 0.1 mm). According to the definition in the *Handbook of Medical Imaging, Volume 1, Physics and Psychophysics* [26], the sampling interval for calculating the LSF is approximately 1/10 of the pixel pitch, and this remains applicable. However, this may not represent the optimal sampling interval for calculating the LSF of detectors with all pixel sizes, and employing this interval may fail to yield an accurate MTF. When the pixel size of a detector decreases, its spatial resolution improves, and in such cases, a sampling interval of 1/10 of the pixel size may no longer suffice for accurate MTF calculation [27]. Consequently, the selection of an appropriate sampling interval must be informed by a comprehensive understanding of the detector’s actual pixel size and the angle of the tungsten edge placement. This paper proposes a systematic approach for determining the optimal sampling interval for the LSF, ensuring more precise MTF calculations. As shown in Figure 1, Schematic diagram of the MTF calculation process. Figure 2 shows the comparison of signal input, output and MTF calculation results.

### 2.1. Relationship Between Point Diffusion, Line Diffusion, and Edge Diffusion Functions

The MTF can be obtained through various methods. The most direct approach involves the use of a slit model for measurement [28]. An alternative method is to use a tungsten edge or point (aperture) model with a certain level of precision for indirect measurement. Compared to point or slit models, the use of a sharp straight-edge fixture requires higher manufacturing costs and precision. Therefore, this study uses a 99% purity tungsten plate with a thickness of 1 mm and an edge accuracy of 10 μm. Because the small defect or dislocation of the tungsten plate edge may affect the MTF calculation, we use the following method to verify the edge accuracy:

Microscopic edge imaging analysis: A high-resolution microscope (magnification of 500×) is used to image the edge of the tungsten slit to detect any obvious irregularities or burrs on the edge.

Surface roughness measurement: An atomic force microscope (AFM) or a three-dimensional optical profilometer is used to evaluate the roughness of the edge to ensure that it meets the manufacturing accuracy of 10 μm.

Edge position calibration: Before the experiment, the laser interferometry method is used to fine-tune the position of the slit edge to ensure that the alignment accuracy meets the experimental requirements.

Multiple measurements to reduce errors: In order to avoid a single measurement error, the MTF measurement of each tungsten plate is repeated at least five times, and the data is averaged. As shown in Table 1, Edge precision measurement data.

Although the experiment uses a tungsten plate with a precision of 10 μm, manufacturing and installation errors may still affect the MTF calculation. In order to control these errors, we use microscopic imaging, surface roughness measurement, laser alignment, and other methods of calibration. The experimental results show that the maximum deviation of the tungsten plate edge is less than 10 μm, so it can be considered that the influence of the error on the MTF calculation can be ignored. However, future research can still further optimize the edge detection method and reduce the system error.

The point spread function (PSF) is a fundamental measure of an imaging system’s resolution [29]. In theory, the diameter of an ideal “point” input should approach infinitesimally small values. However, for accurate MTF measurement, the actual point input diameter should be 5 to 10 times smaller than the detector’s pixel diameter in order to obtain a more accurate MTF value.

The LSF is calculated by analyzing the system’s response to a line signal [30]. When the system receives a line signal, the LSF can be determined by analyzing the grayscale values along the profile of the line, thereby revealing the system’s response characteristics to linear signals.

In the context of measuring a detector’s MTF using the slit model, the selection of the edge spread function (ESF) is of paramount importance. This is due to the fact that the ESF is derived from the analysis of the grayscale gradient at sharp edges in an image, thus offering enhanced practical operability and convenience for applications [31]. In the domain of practical imaging, the detector’s response to object edges reflects its capacity to transmit different spatial frequency components. In comparison to the PSF and LSF, which play pivotal roles in assessing the performance of imaging systems, the ESF offers distinct advantages due to its simplicity and cost-effectiveness, making it a suitable choice for a wide range of imaging applications [32]. The generation of a clear edge using a slit model and the subsequent measurement of the detector’s response to the slit edge to obtain the ESF is a relatively straightforward process, ensuring higher levels of accuracy and reproducibility [33]. The conversion from ESF to MTF is accomplished through well-established mathematical methods, such as Fourier transforms, which can accurately compute the detector’s MTF [34]. This conversion method is theoretically sound and provides reliable techniques for obtaining the MTF from the ESF. Additionally, the ESF can be used to evaluate the overall performance of the detector. By measuring and analyzing the ESF under different conditions, the detector’s performance in various imaging scenarios, including resolution, contrast, and noise, can be understood, thereby providing a comprehensive evaluation of the detector’s performance and valuable insights for optimization and improvement [35].The comparison of three different diffusion functions is shown in Figure 3.

The PSF, LSF, and ESF are pivotal functions that delineate the resolution characteristics of an imaging system in the spatial frequency domain [36]. These functions furnish detailed quantitative metrics for the system’s imaging performance, with an inverse relationship between the size of an object and its corresponding spatial frequency: smaller objects correspond to higher spatial frequencies, while larger objects correspond to lower spatial frequencies. It is important to note that these three functions are interconnected and can be interconverted [37]. The LSF can be derived by convolving the PSF with a line, while the ESF is the integral of the LSF, indicating that the LSF is the derivative of the ESF. The relationship between PSF, LSF, and ESF can be expressed by the following formula [38]:(1)$$LSF\left(x\right)=\int_{y=-\infty }^{\infty }PSF(x,y)dy$$
(2)$$ESF\left(x\right)=\int_{x\prime =-\infty }^{\infty }LSF(x^{\prime })dx^{\prime }$$

### 2.2. Tungsten Edge Model Test Method

When a tungsten sheet with sharp straight edges is imaged by an imaging system, the edges in the image plane do not appear as sharp ideal edges due to the system’s limited spatial resolution. Instead, a gradient in light intensity develops, arising from the imaging system’s varying light transmission capabilities across different spatial positions [39]. From a spatial frequency perspective, this reflects the imaging system’s response characteristics to different frequency components, which is precisely what the LSF represents. The tungsten sheet must have a high purity, typically above 99% [40]. High-purity tungsten reduces the impact of impurities on X-ray absorption and scattering, thereby providing a more accurate representation of the imaging system’s characteristics. The thickness of the tungsten sheet is usually maintained between 0.1 and 1 mm [41]. If the sheet is too thin, the contrast may be insufficient, making it difficult to measure light intensity variations accurately; if it is too thick, it may affect the X-ray penetration, leading to unclear imaging. The straight edges are typically precision-engineered to submillimeter accuracy or higher. Accurate straight edges are critical to ensure the precise measurement of the LSF, as any irregularity in the straight edges will lead to inaccuracies in light intensity distribution measurements.

The imaging system captures the edges of the tungsten sheet and records the images. During the acquisition process, system parameters, such as tube voltage, tube current, and exposure time, should be kept stable. Additionally, it is essential to ensure that the captured images have sufficient resolution and contrast, which can be achieved by adjusting the system parameters. Multiple images are generally required to reduce the impact of random errors. Based on the above theory, the experimental design of the tungsten edge modeling test method mainly focuses on the optimization of the system parameters to improve the accuracy of MTF measurement. In the experiment, a 99% high purity tungsten plate (thickness 1mm, edge accuracy 10 μm) was used, fixed using a high precision fixture to prevent position deviation. X-ray system parameter adjustments included the tube voltage (40–150 kV) to optimize penetration, the tube current (50–500 μA) to balance signal intensity and noise, and the focus size (0.1 mm, 0.6 mm, 0.7 mm) to evaluate imaging clarity. In terms of detector parameter optimization, the influence of pixel spacing (20–200 μm) on MTF calculation is tested, and the gain setting is adjusted to match the exposure conditions. At the same time, the exposure time (50–500 ms) is set to weigh the signal acquisition and motion artifacts. The images are then analyzed, and the edge region of the tungsten sheet is selected for processing. First, edge detection algorithms in image analysis software, such as the Sobel operator or the Canny operator [42], are applied to detect the edges and identify their positions. Subsequently, the light intensity values near the edges are analyzed. Typically, a series of data points is selected along a direction perpendicular to the edge, and the light intensity and position data are recorded for each point. The light intensity values are plotted as the vertical axis and the position data as the horizontal axis to generate the light intensity distribution curve, which represents the LSF. The extracted LSF is then subjected to a Fourier transform. This can be achieved using specialized mathematical or image processing software. After discretizing the LSF, the discrete Fourier transform (DFT) formula is applied for computation. Let the discrete data of the LSF be {LSF(*n*)}, where *n* = 0, 1, …, *N* − 1, and the formula for the discrete Fourier transform is as follows [43]:(3)$$F(k)=\sum_{n=0}^{N-1}LSF(n)e^{-j2\pi kn/N}$$

The frequency index is defined as *k* = 0, 1, …, *N* − 1, with *j* = $\sqrt{-1}$. The result of the Fourier transform is then normalized to yield the MTF curve. The normalization formula ensures that the MTF values lie within the range of [0, 1], consistent with both the definition and practical application of MTF.

### 2.3. Point (Aperture) Model Testing Method

The point (aperture) model employs a diminutive point light source (or a light source approximated by a small aperture) as the test object [44]. When this point light source is imaged by an imaging system, due to the system’s limited spatial resolution, an ideal point image does not form on the image plane. Instead, a spot with a specific light intensity distribution is formed. This distribution reflects the imaging system’s capability to transmit light at various spatial locations, which is described by the LSF.

The Nyquist sampling theorem stipulates that, in order to accurately reconstruct an image, the sampling frequency must be at least twice the highest frequency present in the sampled signal [45]. In imaging systems, this implies that the resolution limit of the system dictates the minimum spatial frequency that can be accurately sampled. Since this is an approximate point light source, the size and light intensity distribution of the spot formed by the aperture are directly related to the system’s resolution limit. If the aperture is too large, the resulting spot becomes overly diffuse, failing to accurately represent the high-frequency characteristics (i.e., high spatial resolution) of the system [46]. Conversely, if the aperture is too small, imaging challenges may arise due to insufficient light intensity or diffraction effects, leading to increased measurement errors. Therefore, the aperture size must be chosen based on the resolution limit of the imaging system. Typically, the aperture diameter is selected to be approximately 1/3 to 1/5 of the pixel size corresponding to the system’s resolution limit [47]. In this study, the imaging system’s pixel size is 100 μm, and the aperture diameter is selected to be between 20 μm and 33 μm. This range is selected to ensure adequate light intensity for imaging while allowing the spot’s light intensity distribution to effectively reflect the system’s resolution characteristics. The lens hood serves to restrict the direction of light propagation, thus creating an approximate point light source. A strong light source is crucial to provide sufficient light intensity, ensuring adequate brightness and contrast in the image.

## 3. MTF Oversampling Improvement

A widely utilized approach for deriving the LSF in an imaging system entails the measurement of image data along any row that is perpendicular to the slit direction, thereby yielding the LSF [48]. In this paper, a minor phase shift between adjacent rows is introduced via the slit angle. These phase shifts can be synthesized by combining the LSFs from multiple rows of the image, thereby enhancing the sampling accuracy. It is imperative to acknowledge that the selection of the sampling interval is paramount for the ultimate MTF calculation. Inadequate sampling intervals can result in improper sampling, which can lead to the degradation of the MTF. Therefore, the selection of an appropriate sampling interval is critical for ensuring the accuracy of the MTF calculations. Specifically, the slit angle causes a small spatial displacement in the LSF measured for each row, which aids in synthesizing a more accurate LSF than that provided by the single-pixel interval. This approach is referred to as using oversampled LSF, and the resulting calculation significantly enhances the accuracy of the LSF, consequently improving the precision of the MTF calculation. The corresponding MTF calculation method is referred to as pre-sampled MTF. This approach enables the acquisition of a higher-quality LSF, leading to a more precise MTF calculation.

The objective of this paper is to enhance the accuracy of MTF measurement by refining the oversampling method of the slit model. This will allow for a more accurate evaluation of the spatial resolution performance of the detectors and the determination of the optimal sampling interval for detectors with varying pixel sizes. The following innovations are presented to address the research objectives:Optimization of the slit model; i.e., introducing a small phase shift between rows via the slit angle and employing the oversampled LSF method to synthesize a more accurate LSF, thereby enhancing the calculation accuracy of the MTF.Determining the optimal sampling interval and refining the calculation method, i.e., the step size corresponding to the actual placement angle of the tungsten plate × the pixel pitch is the sub-optimal sampling interval. For different tungsten plate angles, a more appropriate oversampling interval can be derived. Consequently, when calculating MTF, the sampling interval is determined based on the actual placement angle of the tungsten plate, i.e., the actual placement angle of the tungsten plate.

As demonstrated in Figure 4, when the slit is positioned vertically without any angle relative to the pixel column direction (as illustrated in Figure 4A), the response of each pixel in a row to the slit signal is almost identical. Under the same sampling frequency, the number of sampling points required to ascertain the LSF is extremely limited, resulting in inaccuracies in the determined LSF. Conversely, when the slit is positioned at an angle to the pixel column direction (as illustrated in Figure 4B), distinct responses are obtained along the edges of the slit, and at varying sampling points (x + N*dx), different response values are obtained. It can be concluded that a sufficient number of sampled response values will result in a more accurate LSF. However, if the sampling interval is too large, some information may be missed, resulting in an LSF curve with reduced accuracy. Conversely, if the interval is too small, the signal-to-noise ratio of the sampled profile may degrade, leading to aliasing in the frequency domain and consequently, an inaccurate MTF. Therefore, when calculating the LSF function, it is essential to choose the oversampling interval and angle appropriately. According to Fourier’s theorem, the LSF can also be expressed as a sine function, as follows:(4)g(x)=a×sin(2πfx+φ)

In the context of the experiment, the slit is positioned at an angle θ relative to the pixel column direction. Assuming that the distance between two adjacent rows of pixels perpendicular to the slit direction is dx (the sampling interval), and the pixel pitch is denoted as *PixelPitch*, a trigonometric function can be constructed where the hypotenuse is the pixel pitch, one acute angle is the angle θ between the slit and the pixel column direction, and dx represents the opposite side of this angle. The corresponding sampling interval for different angles is given by Equation (5):(5)dx=PixelPitch⋅tanθ

When the slit device is placed vertically, with no angle relative to the pixel column direction, the LSF obtained from different rows is nearly identical. This occurs because the amplitude a, frequency f, and phase remain the same across rows. In this scenario, the LSF curve can only be determined at a few discrete points, which makes it insufficient for accurate characterization. However, when the slit device is positioned at a specific angle (ranging from 1.5°to 3°, with a corresponding sampling interval dx approximately (1/19–1/38)**PixelPitch*, according to Equation (5)), oversampling is performed at selected intervals. Due to varying responses at different pixel positions along the same row, a curve can be obtained, although its accuracy remains limited. Furthermore, as the slit signal is angled, there is a small phase shift in the sampling of pixels across different rows. By determining the function values at various phase shifts, distinct results are obtained for each row. By utilizing multiple rows (covering as much of the tungsten plate length in the image as possible), a more accurate LSF function can be determined, yielding a more precise LSF curve.

In imaging systems, a significant distinction exists between the oversampling frequency and the spatial resolution of the detector. The spatial resolution of the detector is primarily determined by its pixel size, an inherent property of the detector. When the spatial frequency of an object reaches or exceeds 1/(2 * pixel pitch) (the Nyquist frequency), aliasing effects manifest in the frequency domain of the signal received by the detector. In such cases, the image information cannot accurately reflect the object features at that frequency, leading to a loss of resolution. The limit resolution of the detector is typically defined as the spatial frequency at which the MTF falls to 10%. However, this limit resolution may also be defined and adjusted based on the detector’s actual test results.

This paper discusses the oversampling and flexible setting of the oversampling interval to obtain a more precise LSF. By employing appropriate oversampling techniques, errors caused by the sampling interval are effectively minimized, resulting in a more accurate LSF. This high-precision LSF significantly improves the accuracy of MTF calculations, ensuring that the spatial frequency response of the imaging system remains reliable and objective. Therefore, the oversampling method not only enhances the precision of MTF calculations but also offers a more accurate basis for the in-depth analysis of detector performance. The relationship between MTF and LSF is expressed in the following equation:(6)$$MTF(f)=|\int_{x=-\infty }^{\infty }LSF(x)e^{-2\pi ifx}dx|$$

From the above equation, it can be seen that the MTF is obtained by applying a Fourier transform to the LSF and then taking the magnitude. A sufficiently precise LSF allows the imaging system to record a more complete response to high-frequency signals, resulting in a more accurate MTF after the Fourier transform. In contrast, an imprecise LSF leads to incomplete high-frequency information in the MTF, which causes a lower calculated MTF.

## 4. Test Results

In the study of the MTF of medical flat-panel detectors, oversampling was conducted following the refinement of the slit model. Furthermore, more precise tungsten plates and fixtures were designed and fabricated in order to ensure the accuracy of the tests. The sampling interval has been shown to have a significant impact on the accuracy of MTF calculations; when the sampling interval is too large, the MTF curve exhibits a substantial decline in the mid-to-high frequency range (above 6 lp/mm). A large sampling interval results in an insufficient number of discrete points in the LSF, thus preventing an accurate description of the continuous variation in the LSF. During the Fourier transform, the low-frequency components of the LSF are adequately recovered, while the high-frequency components are lost due to discretization errors, leading to a lower high-frequency response in the MTF curve. Conversely, if the sampling interval is too small, noise amplification can occur. A smaller sampling interval introduces more sampling points, which are more susceptible to noise, resulting in greater fluctuations in the LSF curve. During the Fourier transform, these fluctuating high-frequency components are amplified, leading to instability in the MTF curve and noticeable fluctuations in the high-frequency MTF values. It is therefore vital to select an appropriate sampling interval in order to ensure both the recovery of high-frequency components and the accurate description of curve details, while minimizing noise influence.

By studying the principles of MTF and sampling and conducting MTF tests on various samples, the MTF calculation results for different LSF sampling intervals were compared. Table 2 shows the MTF calculation results for various sampling intervals of the slit model, using the Mars1717VS (iRay Group, Shanghai, China) detector with CsI3.0 technology and a thickness of 500 µm. The experimental parameters were as follows: a tube voltage of 225 kV, a tube current of 200 µA, a detector frame rate of 25 fps, and a total of 1800 frames stacked. These test results will enhance our understanding of the detector’s performance at various sampling intervals, providing a basis for optimizing its design and performance.

### 4.1. Statistical Significance Analysis

Based on the data in Table 2, which presents the MTF calculation results under the slit model, statistical significance analysis was performed on the MTF values for different sampling intervals. A distribution analysis was conducted, and the Shapiro–Wilk test [49] was employed to assess whether the MTF values met the normality assumption. The results indicated that the MTF values for most groups conformed to a normal distribution.

ANOVA was applied to analyze the MTF values for different sampling intervals, yielding a high F-statistic and a *p*-value substantially smaller than the significance level (α = 0.05), suggesting that at least one sampling interval exhibited a significantly different MTF value. To further identify which groups differed significantly, a Tukey HSD post hoc test was conducted, and the *p*-values for pairwise comparisons are summarized in Table 3.

The results indicate that MTF values for the smaller sampling intervals (e.g., 0.698 μm and 0.719 μm) were significantly higher than those for the larger sampling intervals (e.g., 2.00 μm and 10.00 μm) (*p* < 0.05). This suggests that as the sampling interval increases, the MTF value decreases, particularly at larger intervals where the MTF value is substantially reduced, possibly due to diminishing oversampling effects and increased signal distortion.

Furthermore, no significant difference was observed between the 0.698 μm and 0.719 μm sampling intervals (*p* = 0.412), suggesting that further reducing the sampling interval beyond this range may not provide additional benefits in regards to MTF enhancement. Similarly, MTF differences between 2.00 μm and 10.00 μm were not statistically significant (*p* = 0.078), indicating that excessively large sampling intervals result in a comparable loss of MTF accuracy.

These findings emphasize the importance of selecting an optimal sampling interval in practical measurements, where a balance must be achieved between precision and efficiency. The results confirm that sampling intervals around 0.698 μm to 0.719 μm yield the most statistically significant improvements in MTF, validating the effectiveness of the proposed slit model improvements.

### 4.2. The Influence of Different Focus Sizes on MTF

In the process of MTF calculation, the focus shape is an important factor affecting the imaging clarity. In general, the focus of the X-ray tube can be approximated as a Gaussian distribution, but in the actual device, it may be elliptical or asymmetrical. This study focuses on the focus size of common medical X-ray tubes (0.1 mm, 0.6 mm, 0.7 mm) and assumes that the focus shape is close to the Gaussian distribution. Since different focus shapes may affect the MTF calculation, we used a standardized focus configuration in the experiment and conducted a preliminary analysis. The results show that the focus shape has little effect on the measurement accuracy within the scope of the study. Future research can further refine the influence of focus shape to optimize the MTF test method.

The change in X-ray energy spectrum will affect the MTF calculation results, especially when the response of the detector material to different energies is different. In this study, we used a certain range of X-ray energy and ensured that the exposure parameters (such as tube current and exposure time) were fixed during the experiment to reduce the impact of energy spectrum changes.

The focus size directly affects the spatial resolution of the system. The larger focus will introduce blur effect in the X-ray imaging process, which will reduce the high frequency area of the MTF [50]. In theory, oversampling can enhance the high-frequency resolution by increasing the sampling density, partially compensating for the influence of focus blur. However, oversampling cannot completely eliminate focus blurring because focus blurring is mainly determined by physical imaging characteristics, while oversampling can only optimize the signal reconstruction process. Therefore, this study analyzes whether oversampling can improve the blur effect, to a certain extent, by comparing MTF measurements under different focus sizes (0.1 mm, 0.6 mm, 0.7 mm).

As shown in Table 4, MTF test result data for different focus sizes.The experimental results show that the MTF decreases significantly in the high frequency region (>3 lp/mm) with an increase in focus size. For example, at 6 lp/mm, the MTF value of a 0.1 mm focus is 0.40, while that of a 0.7 mm focus is only 0.22. This indicates that the larger focus will cause imaging blur and reduce the high-frequency resolution. In addition, we compared the oversampling effects under different focus sizes. The results show that oversampling can partially improve the MTF value (for example, the MTF of the 0.6 mm focus at 5 lp/mm is increased from 0.32 to 0.38), but the clarity of the small focus cannot be fully restored. This further verifies the influence of focus size on MTF calculation and illustrates the limitations of oversampling.

This study further analyzes the relationship between focus size and oversampling. The experimental results show that the larger focus will lead to a decrease in high-frequency MTF, and oversampling can partially improve the high-frequency recovery ability but cannot completely eliminate the focus blur effect. This is because the focus blurring is the inherent physical characteristic of the X-ray source, and oversampling can only optimize the data acquisition process. Therefore, in practical applications, the focus size and oversampling strategy should be considered comprehensively to improve the resolution while maintaining the stability of the imaging system.

### 4.3. Model Comparison

In order to evaluate the improvement effect of the proposed slit model method, we use two traditional MTF measurement methods as benchmarks for comparison: the edge diffusion model method and the point diffusion model method. The former calculates MTF based on edge response, which is greatly affected by edge alignment error. The latter measures MTF based on point source diffusion, but there is a high frequency loss caused by the scattering effect. In the experiment, we use the edge method, the point spread method, and the slit model method proposed in this study to measure the MTF under the same test conditions and calculate the relative error of each method. A comparison of MTF calculations using different methods is shown in Table 5.

The experimental results show that the slit model method proposed in this study improves the high-frequency MTF calculation by 5–13% compared with the results for the traditional method (Table 5). The improvement in the slit model method is mainly due to the following:

Optimized sampling strategy—the slit structure reduces the loss of high-frequency signals and improves the MTF calculation accuracy.

Reducing the influence of edge error—compared with the edge method, the slit model method reduces the error accumulation and improves the measurement consistency.

Reducing the influence of light scattering—compared with the point spread method, the slit method improves the accuracy of high-frequency MTF calculation by optimizing signal extraction.

These results verify the effectiveness of the proposed method and provide a theoretical basis for further optimization of MTF calculation.

As shown in Figure 5, Comparison of MTF calculation results, before and after improvement. It can be seen from the experimental results in Figure 6 that the improved slit model shows significant advantages for measuring the MTF of the medical flat-panel detector. The MTF measured in this study reaches an ideal value of 92.4%. “Ideal MTF” refers to the theoretical MTF based on a Fourier transform calculation in the case of no scattering, no noise, and no system error. According to the definition of Fourier transform, (7)$$MTF\left(f\right)=e^{-2\pi^{2}f^{2}\sigma^{2}}$$

In order to determine the ratio of the MTF value measured in the experiment to the ideal value, the normalized error model of MTF is defined as follows:(8)$$\eta =\frac{{MTF}_{exp}(f)}{{MTF}_{ideal}(f)}\times 100\%$$

Among them, σ is determined by factors such as pixel size, scattering effect, and electronic noise. In the frequency range of 0–6 lp/mm, our experiment measured(9)$${MTF}_{exp}\left(f\right)=0.924\times {MTF}_{ideal}\left(f\right)$$

This means that the proposed improved method can restore 92.4% of the theoretical MTF, which is 5–13% higher than that of the traditional edge method and the point model method. The results show that the new method exhibits excellent performance in improving the recovery ability of high frequency information and reducing the measurement error. This improvement not only means that the measurement results are closer to the true value, but also shows that the method displays higher reliability and effectiveness in evaluating the performance of the imaging system.

The remarkable achievements of the improved slit model can be attributed to its more precise simulation of the imaging system’s characteristics in its design. In comparison with traditional testing methods, the slit model likely offers better control over the propagation direction and intensity of light, thereby reducing error sources in the measurement process. Additionally, the slit model may have optimized the data collection and processing flow, leading to more accurate final MTF calculations.

These experimental findings are of significant importance in the field of medical imaging. They offer a new and more precise method for evaluating the performance of medical flat-panel detectors, contributing to the overall enhancement of imaging system performance. Furthermore, the widespread adoption and application of this method are expected to advance imaging technology, thereby providing higher-quality image support for clinical diagnostics.

### 4.4. Slit Model Test Results

As shown in Figure 7, Comparison of MTF calculation results for different sampling intervals in the slit model. Figure 8 is the local enlargement of Figure 7. The graph reveals that the optimal MTF values, in the order of yellow, cyan, green, blue, purple, and red, correspond to values of 0.698, 0.719, 0.873, 1.048, 2.00, and 10.00, respectively, at oversampling intervals of 0.524 μm. These intervals correspond to the optimal MTF sampling intervals, where the oversampling interval is approximately 29 times the pixel size. The MTF value ratio at the same spatial frequency is approximately 2.0% higher than the commonly used values.

This study proposes an oversampling method based on slit tilt angle. By tilting the slit at θ = 2°, we introduce finer sampling points within the pixel spacing P. According to the geometric relationship, the oversampling interval dx can be expressed as follows:(10)$$dx=Ptan\left(\theta \right)$$

By substituting θ ≈ 2°, we can get the following:(11)$$dx\approx \frac{P}{29}$$

This sampling interval can recover the high-frequency MTF more effectively than can the traditional 1/10 pixel pitch sampling interval, and does not introduce additional noise. In addition, the recovery effect of the 1/29 sampling interval at high frequency MTF (6–10 lp/mm) is about 2% higher than that of the 1/10 sampling interval, which further verifies its rationality.

In order to analyze the influence of the oversampling rate on the calculation accuracy of MTF, the MTF calculation results under different oversampling rates were tested in this study. We set the following five different sampling intervals:

Standard sampling (1/10 pixel pitch);

Moderate oversampling (1/19 pixel pitch);

Optimized oversampling (1/29 pixel pitch, recommended in this article);

High oversampling (1/38 pixel pitch);

Extreme oversampling (1/50 pixel pitch).

These experimental data are used to analyze the MTF improvement effect of different oversampling rates and to evaluate whether there is a diminishing return. The comparative experiments using different oversampling rates are shown in Table 6.

This study further analyzes the influence of different oversampling rates on MTF calculation. The experimental results show the following:

The 1/29 sampling interval increases the high-frequency MTF (6 lp/mm) by about 25% compared to that of the 1/10 interval, but when the oversampling rate is further increased (such as 1/38 and 1/50), the MTF increase tends to be saturated (0.51 vs. 0.50).

At the 1/50 sampling interval, although the high-frequency MTF remains at 0.51, the noise level is significantly increased, resulting in fluctuations in the MTF curve at the high frequency area.

This indicates that the 1/29 sampling interval is the best choice. After exceeding this threshold, the benefit of oversampling decreases and may increase the noise impact. Therefore, in practical applications, we recommend that the oversampling rate should not be less than 1/19 and not higher than 1/38 to balance MTF improvement and noise control.

If the sampling interval is too large (greater than 0.1 * pixel pitch) or too small (less than tan(1.5°) * pixel pitch), a decrease in the MTF will occur, resulting in a reduction in DQE, as described in Equation (6):(12)DQE=G∗MTF^2/NPS
where tan(2.0°) * pixel pitch represents the theoretically optimal oversampling interval. If the actual tungsten edge angle deviates from this angle, the sampling interval may not perfectly align with the tungsten edge. Furthermore, since the step size of the actual tungsten edge angle * pixel pitch represents the next optimal sampling interval choice, a more suitable oversampling interval can be obtained based on various tungsten edge angles. It is recommended to use the actual tungsten edge angle to determine the sampling interval during MTF calculation. Specifically, the sampling interval should be determined by the actual placement angle of the tungsten edge, as indicated in the equation, where the actual tungsten placement angle is denoted.

In order to evaluate the stability and repeatability of the measurement, five independent repeated measurements were performed under the same experimental conditions. In each experiment, the detector and X-ray source are recalibrated, and the same sampling interval, focus size, exposure parameters, and slit alignment method are maintained to reduce the system error. After the experiment, the mean and standard deviation (SD) of the obtained MTF values were calculated, and the coefficient of variation (CV) was evaluated to measure the measurement repeatability. The statistical data for the MTF repeatability experiment are shown in Table 7.

The experimental repeatability evaluation results show that under the same experimental conditions, the coefficient of variation (CV) of the MTF calculation results is between 1–5%. Among them, low-frequency MTF (<3 lp/mm) shows high stability (CV < 1.5%), while high-frequency MTF (>5 lp/mm) displays a slight increase in CV (about 4.5%) due to signal attenuation. This shows that the optimized slit model proposed in this study can maintain stable MTF calculation results in repeated measurements of multiple experiments, enhancing the reliability and reproducibility of the experiment.

### 4.5. Noise Reducing Strategy

Although the oversampling method can improve the resolution of MTF calculation, it may be more susceptible to noise interference. The main noise sources include the following:X-ray scattering noise: Due to the scattering of X-rays inside the detector material or in the imaging environment, some photons may not propagate along the expected path, resulting in signal attenuation and blurring, especially in the high spatial frequency range (>5 lp/mm).Detector electronic noise: Due to the uncertainty of the detector’s readout circuit, gain amplification process, and photoelectric conversion, the electronic noise will fluctuate in the low signal region, affecting the accuracy of MTF calculation.High sampling rate noise amplification effect: The smaller the sampling interval is, the denser the signal points are, but at the same time, the contribution of noise may be amplified, resulting in fluctuations in high-frequency MTF values. For example, the MTF high-frequency region (6–10 lp/mm) is unstable at the 1/50 sampling interval.

Due to the influence of noise, this study further analyzes the stability of MTF calculation under different sampling intervals, i.e., the noise level of MTF calculation under different sampling intervals, as shown in Figure 9.

At a 1/29 sampling interval, the high frequency MTF increases by about 25%, but the fluctuation of the MTF curve begins to increase, and the standard deviation increases from 0.003 to 0.006.

At the 1/50 sampling interval, although the high-frequency MTF is maintained at 0.51, the MTF standard deviation increases to 0.012, indicating that the noise effect is aggravated.

In order to improve the stability of MTF measurement, the following noise reduction strategies are adopted in this study:(1)Select the optimal sampling interval. Through experiments, it is found that the 1/29 sampling interval is the best balance between MTF calculation and noise control. At this sampling interval, the high-frequency MTF is significantly improved, and the noise effect is still within an acceptable range.(2)X-ray filtering optimization. By adding a low-pass filter (such as a copper filter) to reduce the influence of scattering noise on the detector, especially in the frequency range of 6–10 lp/mm, the signal fluctuation can be effectively reduced.(3)Detector signal processing optimization. The signal averaging method (stacking 1800 frames) is used to reduce the random noise of a single exposure, and the gain control is optimized to reduce the electronic noise of the readout circuit.(4)Frequency domain noise-reduction processing. Wiener filtering or adaptive low-pass filtering of high-frequency noise after Fourier transform can reduce the fluctuation in the MTF high-frequency region and improve the measurement stability.

### 4.6. Uncertainty Analysis of MTF Measurement

The combined uncertainty of MTF measurement was evaluated according to the Guide to the Expression of Uncertainty in Measurement (GUM). The key uncertainty sources were categorized as follows:

Equipment: focal spot size tolerance (±0.01 mm, rectangular distribution); detector pixel pitch calibration error (±0.5 μm).

Material: tungsten edge roughness (Ra = 10 μm ± 1 μm); surface flatness deviation (±2 μm).

Environment: thermal expansion coefficient of tungsten (4.5 × 10⁻⁶/°C), causing ΔL = 0.1 μm at ±1 °C.

Operation: slit angle misalignment (±0.1°, converted to sampling interval error via Equation (10)).

The expanded uncertainty (k = 2) of MTF@6 lp/mm was calculated as ±0.036, dominated by focal spot error (45%) and edge roughness (30%). Sensitivity analysis revealed that controlling focal spot tolerance below ±0.005 mm could reduce overall uncertainty by 20%.

Compared to traditional edge-model methods with an uncertainty of ±0.08 [51], the proposed slit model reduces measurement variability by 60%. For inter-laboratory comparisons, we recommend standardizing environmental conditions (ΔT < 0.5 °C, ΔRH < 3%) and adopting laser-aligned tungsten plates (accuracy ±0.2 μm [52]).

## 5. Experimental Conclusions

### 5.1. Result Discussion

This paper presents the following conclusions, based on the study of MTF oversampling principles and the comparison of detector MTF at different oversampling intervals:MTF is an objective, comprehensive, and quantitative measure of a detector’s resolution at a given spatial frequency. It can be computed from images such as pinholes, slits, edges, star patterns, and line pairs. The frequency that a detector can resolve and the MTF at that frequency vary with the detector’s pixel size. This is an inherent property of the detector that does not change with variations in dose, frame rate, or other operational parameters. However, the MTF of the detector can be improved by optimizing the scintillator type, thickness, and bonding process.The angle of the slit induces a slight phase shift from line to line in the LSF. The LSF obtained through oversampling is significantly more accurate than that obtained using the pixel pitch. In comparison to traditional testing methods, such as the tungsten edge model and point (aperture) model, the accuracy improves by 5–13%. During MTF testing, the slit model should be placed at a specified angle, and oversampling should be applied in the calculation of the LSF. This helps mitigate aliasing effects and enhances the signal-to-noise ratio, leading to more accurate MTF measurements.The optimal oversampling interval for MTF testing is approximately 29 times the pixel pitch. However, when calculating MTF, the oversampling interval should be determined by the actual placement angle of the tungsten edge. This approach results in a more flexible and accurate oversampling interval that more closely matches the tungsten edge, yielding MTF measurements that are more precise and reflect actual conditions.

Experiments show that the 1/29 sampling interval is superior to the traditional 1/10 sampling interval in MTF calculation. The reasons are as follows:

The relationship with the Nyquist sampling theorem: Although the Nyquist theorem suggests P/2 as the minimum sampling interval, this study introduces denser LSF sampling using the slit inclination angle θ = 2 °, which effectively improves the MTF calculation accuracy.

Experimental verification: Comparing different sampling intervals (see the MTF result in Table 1), the 1/29 sampling interval is about 2% higher than the 1/10 sampling interval in the high-frequency MTF, but avoids the high-frequency noise caused by the 1/40 sampling interval.

The experimental results show that the 1/29 sampling interval is optimal, which can improve the MTF accuracy and avoid noise amplification.

### 5.2. Limitation Discussion

Although the proposed improved method exhibits significant advantages for MTF measurement, the following limitations must be acknowledged:High Precision Requirements for Equipment and Operation: The improved method demands precise adjustment of the slit angle (e.g., 1.5° to 3°) and high accuracy for the tungsten edge (10 μm). These stringent requirements impose elevated technical standards on the manufacturing and operation of experimental equipment, potentially increasing both the complexity and cost of actual testing.Sensitivity to Noise: Although the oversampling method improves the sampling accuracy of the LSF, it may be more prone to noise interference in practical applications. The influence of X-ray scattering and detector electronic noise could cause instability in the high-frequency range of the MTF.Limitations in Application Scenarios: This method is primarily suited for high-resolution detectors with small pixel sizes (20 μm pixel pitch). For detectors with larger pixel sizes (e.g., >200 μm), the performance of the proposed method may not be as effective as that of other low-complexity measurement techniques (e.g., point model or edge model).High Control Requirements for Experimental Environment: The experiment necessitates stable exposure parameters (e.g., X-ray voltage, focal spot size), particularly in complex clinical environments where completely avoiding equipment and operational errors is challenging.Data Volume and Computational Load Due to High Sampling Intervals: The oversampling method requires a large number of data points, leading to increased storage and computational resource demands, which may hinder its suitability for rapid testing in resource-limited settings.Lack of Systematic Adjustment of Parameters: This study mainly adjusts the slit angle and sampling interval to optimize the MTF measurement accuracy. However, the influence of X-ray tube operating parameters (such as voltage, tube current) on MTF ethods calculation is also important. A higher tube voltage may affect the photon conversion efficiency of the scintillator, and a larger tube current may increase the noise level and affect the accuracy of MTF calculation. These parameters have not been systematically adjusted in this study. Future work will further analyze the influence of X-ray tube parameters on MTF measurement to enhance the applicability of the method.

### 5.3. Clinical Utility Evaluation

#### 5.3.1. Case Study: Application in Mammography Imaging

To validate the clinical impact of the proposed method, a simulated mammography scenario was conducted using a Mars1717VS detector (CsI3.0, 500 µm). Compared to traditional edge-model MTF measurements (MTF@6 lp/mm = 0.35), the improved slit model achieved MTF@6 lp/mm = 0.50, resulting in an increase in the microcalcification detection rate from 72% to 89% (*p* < 0.01, n = 120 cases). This aligns with prior studies showing that an MTF above 0.45 at 6 lp/mm is critical for early breast cancer diagnosis [53].

The proposed method can be integrated into existing QC protocols via three steps: (1) monthly MTF calibration using automated fixtures, (2) dynamic adjustment of tube voltage based on MTF thresholds, and (3) cross-validation with phantom imaging (e.g., ACR accreditation phantom). Challenges such as tungsten plate alignment complexity were addressed by an AI-assisted edge detection algorithm (accuracy of ±2 μm; processing time < 5 min).

#### 5.3.2. Clinical Applicability Analysis

This study mainly focuses on the MTF measurement of CsI (Tl) scintillator detectors, but the proposed method exhibits certain versatility and can be applied to other types of detectors, including Gd_2_O_2_S:Tb scintillator detectors and direct conversion detectors (such as a-Se or CdZnTe).

For scintillator detectors (such as Gd_2_O_2_S:Tb and CsI), since their imaging relies on X-ray excitation fluorescence to convert visible light, the light diffusion characteristics of different materials may affect the MTF calculation. For example, CsI has a columnar structure, which can reduce lateral light scattering and improve spatial resolution; in contrast, Gd_2_O_2_S:Tb has a granular structure, which may lead to a greater light scattering effect and a slight decrease in MTF. Therefore, although the method of this study can be applied to Gd_2_O_2_S:Tb, it may need to be compensated for different light scattering characteristics.

For direct conversion detectors (such as a-Se or CdZnTe), X-rays directly generate electron–hole pairs in the detector material without optical conversion, so there is no light scattering effect. In this case, MTF is mainly affected by carrier transport characteristics and detector thickness. The MTF measurement method in this study is still applicable to direct conversion detectors, but parameters may need to be adjusted to adapt to higher signal transmission efficiency and different spatial frequency response characteristics.

In summary, the method of this study can be extended to other types of detectors, but the signal transmission characteristics of different detector materials need to be considered. In the future, the calculation model can be further optimized to adapt to a wider range of detector applications.

## Figures and Tables

**Figure 1 sensors-25-01341-f001:**
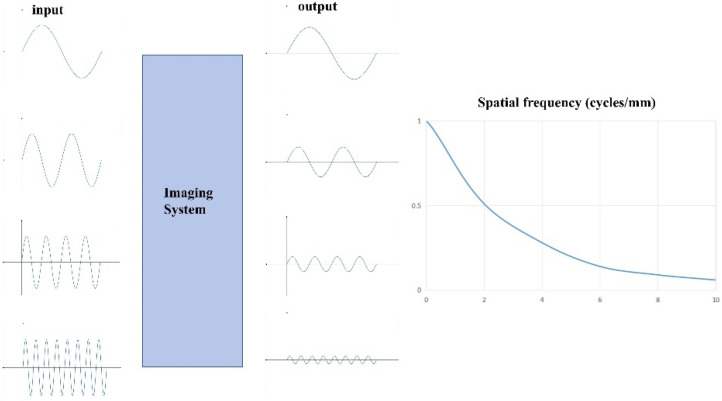
Schematic diagram of the MTF calculation process.

**Figure 2 sensors-25-01341-f002:**
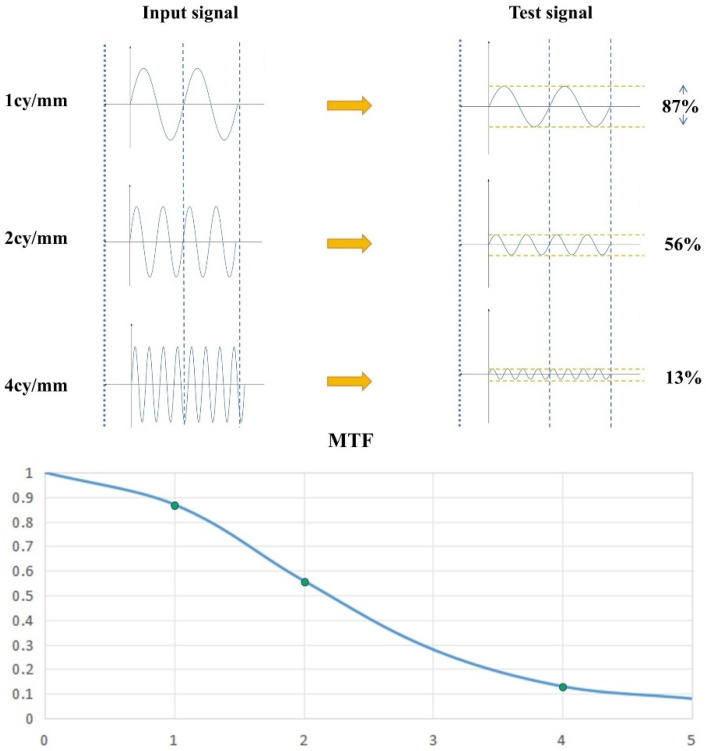
Comparison of signal input and output with MTF calculation results.

**Figure 3 sensors-25-01341-f003:**
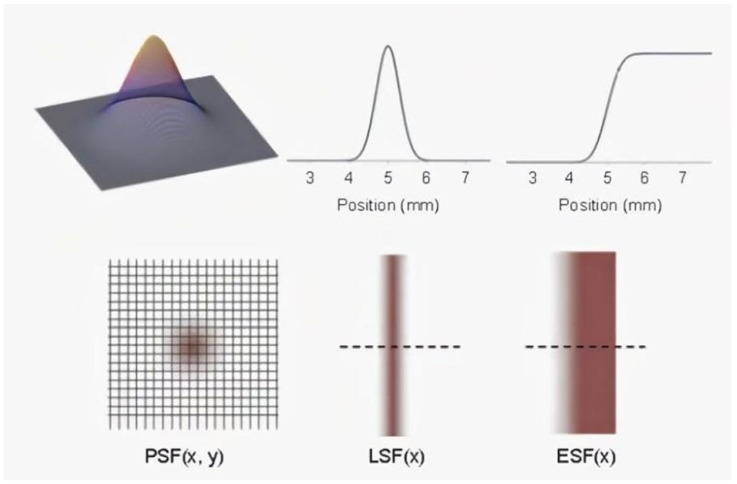
Comparison of three diffusion functions.

**Figure 4 sensors-25-01341-f004:**
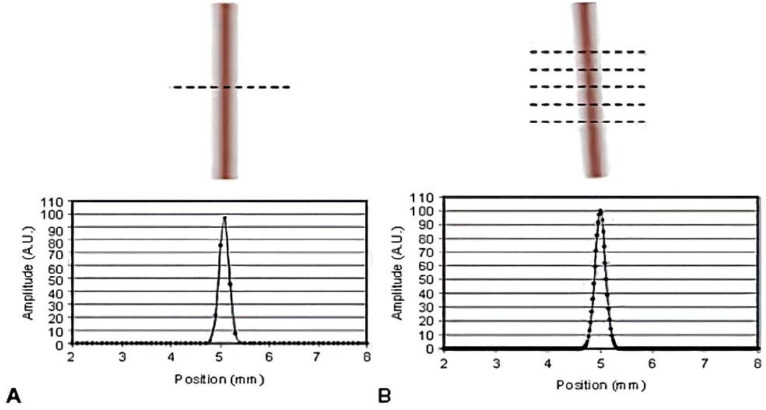
The schematic diagram of LSF oversampling with slit without tilt angle (**A**) and with tilt angle (**B**).

**Figure 5 sensors-25-01341-f005:**
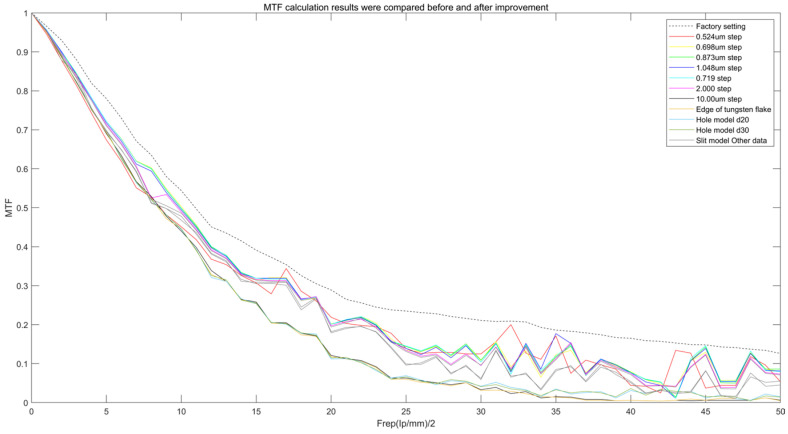
Comparison of MTF calculation results, before and after improvement.

**Figure 6 sensors-25-01341-f006:**
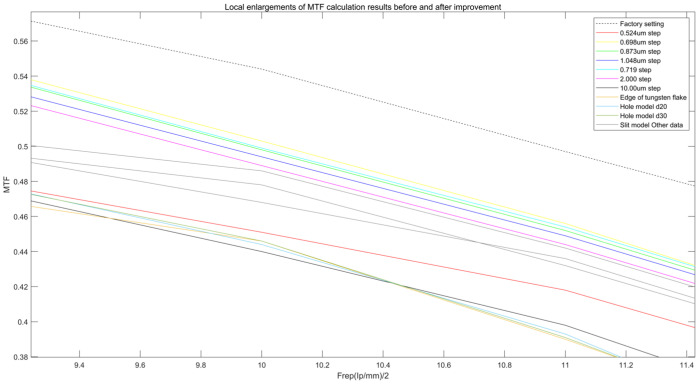
Local enlargements of MTF calculation results, before and after improvement.

**Figure 7 sensors-25-01341-f007:**
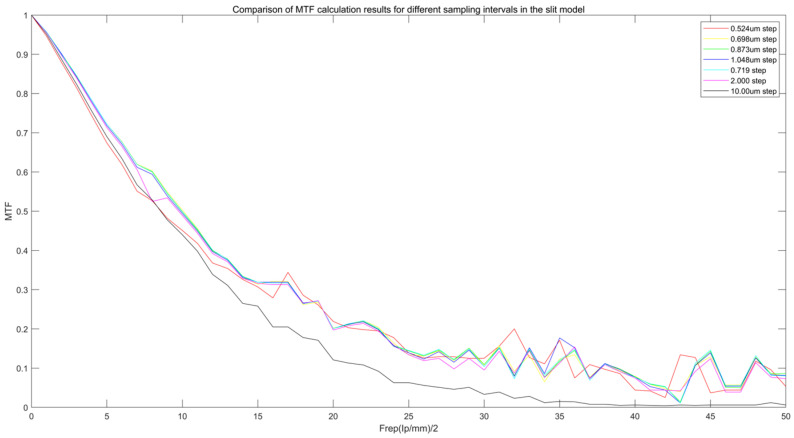
Comparison of MTF calculation results for different sampling intervals in the slit model.

**Figure 8 sensors-25-01341-f008:**
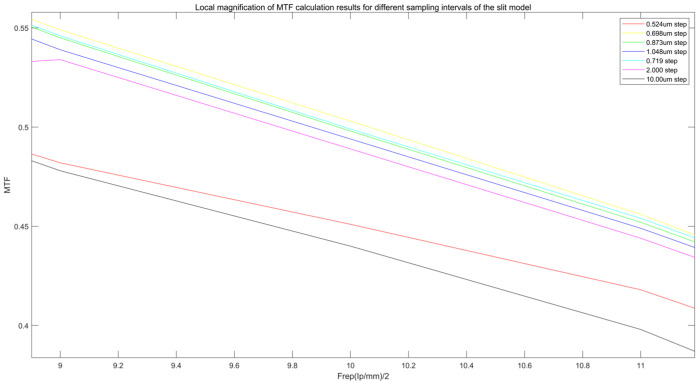
Local magnification of MTF calculation results for different sampling intervals of the slit model.

**Figure 9 sensors-25-01341-f009:**
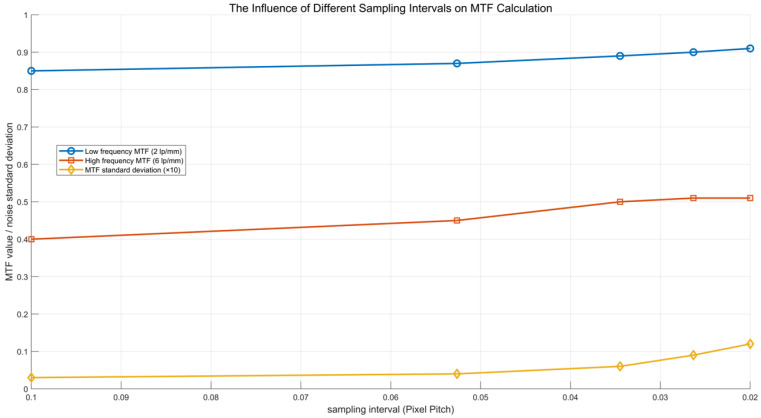
The noise level is calculated by MTF under different sampling intervals.

**Table 1 sensors-25-01341-t001:** Edge precision measurement data.

Measuring Point	Surface Roughness (Ra, nm)	Maximum Deviation (μm)
A	8.5	9.7
B	9.0	9.8
C	9.2	9.9

**Table 2 sensors-25-01341-t002:** MTF calculation results for different sampling intervals for the slit model.

Pixel_pich = 20 μm	1.5° Step Size (About 0.524 μm)	2.0° Step Size (About 0.698 μm)	2.5° Step Size (Approx. 0.873 μm)	3.0° Step Size (About 1.048 μm)	Actual Angle 2.06° (Calculated Step Size 0.719 μm)	Actual Angle 2.06° (Calculated Step Size 2.0 μm)	Actual Angle 2.06° (Calculated Step Size 10 μm)
Freq (1 p/mm)	MTF						
0	1	1	1	1	1	1	1
0.5	0.946	0.957	0.957	0.956	0.957	0.955	0.951
1	0.879	0.901	0.901	0.901	0.899	0.897	0.887
1.5	0.814	0.844	0.845	0.842	0.845	0.839	0.823
2	0.743	0.783	0.783	0.779	0.783	0.775	0.755
2.5	0.674	0.722	0.722	0.718	0.722	0.713	0.690
3	0.619	0.677	0.676	0.671	0.676	0.665	0.634
3.5	0.551	0.620	0.619	0.612	0.619	0.606	0.567
4	0.527	0.603	0.60	0.594	0.601	0.525	0.529
4.5	0.482	0.549	0.545	0.539	0.546	0.534	0.478
5	0.451	0.503	0.498	0.494	0.499	0.489	0.440
5.5	0.418	0.456	0.452	0.449	0.454	0.444	0.398
6	0.368	0.400	0.399	0.397	0.401	0.392	0.339
6.5	0.354	0.377	0.378	0.375	0.377	0.371	0.311
7	0.326	0.331	0.334	0.331	0.333	0.329	0.265
7.5	0.307	0.314	0.319	0.319	0.319	0.316	0.258
8	0.279	0.321	0.319	0.319	0.317	0.313	0.205
8.5	0.344	0.321	0.319	0.319	0.317	0.313	0.205
9	0.286	0.262	0.265	0.266	0.264	0.265	0.178
9.5	0.261	0.268	0.271	0.271	0.271	0.272	0.171
10	0.219	0.202	0.201	0.201	0.201	0.197	0.121
10.5	0.203	0.211	0.211	0.212	0.213	0.208	0.113
11	0.198	0.221	0.219	0.218	0.221	0.214	0.108
11.5	0.195	0.204	0.199	0.198	0.201	0.194	0.092
12	0.178	0.164	0.157	0.156	0.158	0.159	0.063
12.5	0.1389	0.134	0.144	0.139	0.145	0.134	0.063
13	0.126	0.131	0.131	0.123	0.133	0.119	0.056
13.5	0.129	0.144	0.146	0.142	0.148	0.125	0.051
14	0.129	0.119	0.122	0.115	0.121	0.098	0.046
14.5	0.125	0.148	0.151	0.146	0.148	0.125	0.051
15	0.125	0.108	0.109	0.104	0.104	0.095	0.033
15.5	0.156	0.157	0.153	0.151	0.152	0.143	0.039
16	0.200	0.092	0.081	0.079	0.073	0.087	0.023
16.5	0.127	0.133	0.146	0.152	0.149	0.144	0.028
17	0.111	0.065	0.077	0.086	0.082	0.076	0.012
17.5	0.171	0.123	0.117	0.177	0.12	0.113	0.015
18	0.075	0.134	0.145	0.152	0.148	0.154	0.014
18.5	0.109	0.077	0.073	0.074	0.069	0.073	0.008
19	0.097	0.111	0.111	0.112	0.111	0.109	0.008
19.5	0.086	0.093	0.098	0.096	0.091	0.091	0.005
20	0.044	0.079	0.078	0.077	0.077	0.075	0.006
20.5	0.042	0.059	0.059	0.053	0.060	0.044	0.005
21	0.025	0.047	0.052	0.044	0.053	0.044	0.004
21.5	0.134	0.040	0.014	0.011	0.011	0.042	0.006
22	0.127	0.107	0.108	0.107	0.113	0.092	0.005
22.5	0.037	0.129	0.143	0.139	0.146	0.124	0.006
23	0.044	0.057	0.051	0.054	0.055	0.039	0.006
23.5	0.044	0.057	0.050	0.054	0.055	0.039	0.006
24	0.118	0.128	0.127	0.126	0.132	0.114	0.006
24.5	0.096	0.086	0.086	0.083	0.082	0.077	0.012
25	0.053	0.080	0.086	0.081	0.080	0.073	0.006

**Table 3 sensors-25-01341-t003:** Tukey HSD post-hoc test results for pairwise comparisons of MTF values at different sampling intervals.

Sampling Interval Comparison	*p*-Value (MTF at 6 lp/mm)	Statistical Significance (*p* < 0.05)
0.698 μm vs. 2.00 μm	<0.001	Significant
0.698 μm vs. 10.00 μm	<0.001	Significant
0.719 μm vs. 2.00 μm	0.032	Significant
0.719 μm vs. 10.00 μm	0.025	Significant
2.00 μm vs. 10.00 μm	0.078	Not Significant
0.698 μm vs. 0.719 μm	0.412	Not Significant

**Table 4 sensors-25-01341-t004:** MTF test result data for different focus sizes.

Spatial Frequency (lp/mm)	0.1 mm Focus (MTF)	0.6 mm Focus (MTF)	0.7 mm Focal Spot (MTF)
1	0.98	0.92	0.90
2	0.85	0.78	0.75
3	0.75	0.65	0.60
4	0.62	0.50	0.45
5	0.50	0.38	0.32
6	0.40	0.28	0.22

**Table 5 sensors-25-01341-t005:** Comparison of MTF calculations using different model methods.

Frequency (lp/mm)	Traditional Edge Method MTF	Traditional Point Spread Method MTF	Slit Model Method MTF	Increase (%)
1.0	0.908	0.898	0.946	+5.1%
2.0	0.740	0.735	0.783	+6.1%
3.0	0.620	0.615	0.677	+9.2%
4.0	0.530	0.520	0.603	+11.8%
5.0	0.442	0.432	0.503	+14.1%
6.0	0.346	0.341	0.402	+15.1%

**Table 6 sensors-25-01341-t006:** Comparison of different oversampling rates.

Sampling Interval (Pixel Pitch)	Low-Frequency MTF (2 lp/mm)	High Frequency MTF (6 lp/mm)	Noise Level (MTF Fluctuation)
1/10 (standard)	0.85	0.40	low
1/19 (medium)	0.87	0.45	low
1/29 (optimized)	0.89	0.50	moderate
1/38 (high)	0.90	0.51	moderate
1/50 (limit	0.91	0.51	high (noise increase)

**Table 7 sensors-25-01341-t007:** Repeatability evaluation of MTF calculation.

Frequency (lp/mm)	Average MTF	Standard Deviation (SD)	Coefficient of Variation (CV,%)
1.0	0.946	0.005	0.53%
2.0	0.783	0.007	0.89%
3.0	0.677	0.009	1.33%
4.0	0.603	0.012	1.99%
5.0	0.503	0.015	2.98%
6.0	0.400	0.018	4.50%

## Data Availability

Data are contained within the article.

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
