# Peer review of "Improved MTF Measurement of Medical Flat-Panel Detectors Based on a Slit Model"

_sensors, 2025, doi:10.3390/s25051341_

Round 1

Reviewer 1 Report

Comments and Suggestions for Authors

This paper focuses on the measurement of the modulation transfer function (MTF) of medical flat-panel detectors. The topic is of great practical significance, the research methods are systematic, and the experimental design is reasonable. The paper has made contributions to improving the accuracy of MTF measurement. However, there is still room for improvement:

1. The experiments only tested specific detector models (such as Mars1717VS) and a limited range of experimental parameters. Extensive experiments on detectors of different brands and technical types were not carried out, making it impossible to fully verify the universality of the improved method on various detectors. Moreover, the experimental parameter settings are relatively limited. For example, only the slit angle and sampling interval were changed, and the effectiveness of the method under changes in other key parameters such as X-ray tube voltage and current was not explored.

2. The paper points out that the oversampling method may be more susceptible to noise interference, but the analysis of noise sources, impact mechanisms, and quantitative analysis is insufficient. The specific impact degree of X-ray scattering, detector electronic noise, etc. on MTF measurement under different experimental conditions was not studied in detail. Also, effective noise reduction strategies or correction methods were not proposed, affecting the reliability and stability of the measurement results.

3. In the part of clinical application value, the significance of the improved method for imaging quality assessment and detector development was only theoretically expounded, but it was not verified by combining actual clinical cases or simulated clinical scenarios. How this method can be integrated into existing clinical workflows and the challenges and solutions that may be faced in actual operations were not discussed, reducing the clinical transformation potential of the research results.

4. A comprehensive evaluation of the uncertainty of MTF measurement is lacking. There are various influencing factors in the measurement process, such as tungsten plate manufacturing tolerances, equipment measurement errors, and environmental factor fluctuations. Ignoring the uncertainty analysis makes it difficult to accurately judge the accuracy and reliability of the measurement results, and is also not conducive to the comparison and communication of measurement results among different laboratories.

5.Some reports related to this work may should be included in the references. Such as Sensors & Actuators: B. Chemical 326 (2021) 128991; Advanced Science 9 (2022) 2202505; Chemical Engineering Journal 498 (2024) 155355

Comments on the Quality of English Language

The English could be improved to more clearly express the research.

Reviewer 2 Report

Comments and Suggestions for Authors

In this paper, the authors proposed an enhanced MTF oversampling method based on the slit model, which was developed through the evaluation and application of medical flat panel detectors. This method was demonstrated to simulate the characteristics of the imaging system more accurately during the design phase and offers a more precise approach for assessing detector performance. This work is important, at least in the viewpoint of this reviewer, in the field of detectors. It is thereof recommended to publish after considering following comments:

[1]. Please provide a detailed explanation of the experimental design used to adjust the system parameters in the tungsten edge modeling test method described in section 2.2.

[2]. What does Figure 2 illustrate regarding the comparison of signal input and output, along with the MTF calculation results? What is the primary objective of this comparison?

[3]. What criteria are used to select an appropriate sampling interval, as mentioned in section 3 on MTF oversampling improvement?

[4]. Please explain the reason for the significant difference between the MTF values for the smaller sampling intervals (0.698 μm and 0.719 μm) and the MTF values for the larger sampling intervals (2.00 μm and 10.00 μm) in Table 1.

Reviewer 3 Report

Comments and Suggestions for Authors

This paper presents valuable research, but it requires major clarifications and refinements in methodology, experimental justification, and statistical rigor. The following revisions should be addressed before resubmission:

The manuscript states that the proposed slit model approach improves accuracy by 5-13%, but it lacks a clear benchmark for this claim. The reference point for this improvement must be explicitly stated—whether it is based on prior experimental studies, theoretical predictions, or comparisons with conventional MTF measurement methods.

Two additional works should be added to the literature review to strengthen the discussion on medical device innovation and technological advancements in imaging technology. The first study explores innovation strategies in medical devices, while the second examines technological efficiency in medical manufacturing, both providing relevant context for MTF measurement improvements.

*Medical device product innovation choices in Asia: an empirical analysis based on product space.

*How can China's medical manufacturing listed firms improve their technological innovation efficiency?

The ideal MTF value of 92.4% is reported, but there is no clear explanation of how this value was determined. If this is derived from theoretical calculations, a supporting reference or equation should be included. If it is based on experimental data, a justification for why this value represents the ideal condition is necessary.

The study discusses the impact of focal spot size variations (0.1 mm, 0.6 mm, and 0.7 mm), but it does not analyze the effect of focal spot shape or X-ray energy spectrum on MTF calculations. Since focal spot characteristics influence image sharpness, further clarification is needed on whether these factors were controlled or accounted for in the study.

The relationship between focal spot size and oversampling is not clearly defined. If oversampling compensates for the blur caused by larger focal spots, this should be demonstrated through comparative MTF results for different focal spot sizes.

The manuscript assumes perfect edge alignment and sharpness in the tungsten slit, but minor imperfections or misalignments could introduce systematic errors. Since the tungsten plate is described as having a 10 μm edge precision, the verification method for this precision should be described. A supporting measurement, such as microscopic edge imaging or surface roughness analysis, would strengthen the claim.

The derivation of the 1/29 pixel pitch sampling interval for oversampling is unclear. While the results show its effectiveness, the mathematical or experimental basis for this choice is not explicitly discussed. A step-by-step justification is required, particularly since the Nyquist theorem would typically suggest an oversampling rate based on 1/2 pixel pitch constraints.

The paper presents oversampling as a way to improve measurement accuracy but does not evaluate whether increasing the oversampling rate beyond a certain threshold yields diminishing returns. A brief comparison of MTF accuracy at different oversampling rates could confirm whether this method has an optimal range beyond which further refinement is unnecessary.

The ANOVA analysis lacks specific details on which groups showed statistically significant differences. The manuscript should include a summary table with p-values for pairwise comparisons to clarify where the differences are most pronounced.

The paper does not provide an assessment of measurement repeatability under identical experimental conditions. Since MTF is highly sensitive to slight variations in test setup, a brief evaluation of inter-trial variability would add credibility to the reported improvements.

The study focuses on a CsI detector, but the performance of MTF measurement methods can vary across different scintillator or direct-conversion detectors. A short discussion on whether this method is applicable to other detector materials would improve the generalizability of the findings.

Round 2

Reviewer 1 Report

Comments and Suggestions for Authors

The authors have made good improvements to the manuscript, which I think will be accepted for publication.